| **Editor's Pick** | Physiology and Metabolism | Research Article

# Metabolic response of a chemolithoautotrophic archaeon to carbon limitation

Logan H. Hodgskiss,[1] Melina Kerou,[1] Zhen-Hao Luo,[1] Barbara Bayer,[2,3] Andreas Maier,[4] Wolfram Weckwerth,[5,6] Thomas Nägele,[7] Christa Schleper[1]

**ABSTRACT** The ubiquitously distributed ammonia-oxidizing archaea generate energy from ammonia and build cell mass from inorganic carbon sources, thereby contributing to both the global nitrogen and carbon cycles. However, little is known about the regulation of their predicted core carbon metabolism. A thermodynamic model for *Nitrososphaera viennensis* was developed to estimate the consumption of inorganic carbon in relation to ammonia consumed for energy and was tested experimentally by growing cells in carbon-limited and excess conditions. A combined proteomic and metabolomic approach to the experimental conditions revealed distinct metabolic adaptation depending on the amount of carbon supplied, either in a catalase or pyruvate background as a reactive oxygen species scavenger. Integration of protein and metabolite dynamics revealed a cellular strategy under carbon limitation to maintain a pool of amino acids and an upregulation of proteins necessary for translation initiation to stay primed for protein synthesis. The combination of modeling and functional genomics fills gaps in the understanding of the central metabolism and its regulation in a chemolithoautotrophic, ammonia-oxidizing archaeon, even in the absence of available genetic tools.

**IMPORTANCE** Little is known about the regulation of carbon metabolism within ammonia-oxidizing archaea (AOA), a widespread clade that plays a critical role in the global nitrogen cycle while also fixing inorganic carbon. To address this missing knowledge, the soil AOA *Nitrososphaera viennensis* was subjected to various levels of inorganic carbon and analyzed via a systems biology approach to better understand how its core metabolism is regulated. The results demonstrate a strong dependence on the carbon fixation cycle and highlight key connection points between the core metabolic pathways. The analysis additionally revealed tight control on translational processes and elucidated unique cellular responses when the organism was exposed to either exogenous catalase or pyruvate to relieve oxidative stress from reactive oxygen species. The presented data highlight metabolic responses of *N. viennensis* and provide a better understanding of how the organism, and likely other AOA, respond to various environmental conditions.

**KEYWORDS** ammonia oxidation, archaea, carbon limitation, nitrogen cycle, oxidative stress, nitrification

Ammonia-oxidizing archaea (AOA) have been found in a wide variety of environments, often outnumbering their bacterial counterparts, ammonia-oxidizing bacteria (AOB) (1–7). Based on their abundance, their contributions to the global nitrogen cycle are substantial. There have been many efforts to unravel the archaeal ammonia oxidation pathway (8–13). However, their carbon metabolism also draws

Address correspondence to Logan H. Hodgskiss, logan.hodgskiss@univie.ac.at, or Thomas Nägele, thomas.naegele@lmu.de.

The authors declare no conflict of interest.

See the funding table on p. 17.

attention, as all characterized AOA thus far are autotrophic and therefore represent a direct biological link between the global nitrogen and carbon cycles.

AOA participate in the fixation of inorganic carbon by a unique version of the 3-hydroxypropionate/4-hydroxybutyrate (3-HP/4-HB) cycle. Although it appears similar to the 3-hydroxypropionate/4-hydroxybutyrate cycle in hyperthermophilic archaea (14), it is more efficient due to the use of only one ATP equivalent (rather than two) in selected enzymatic reactions and the use of promiscuous enzymes that can catalyze multiple steps in the pathway (14, 15). Most of the biochemical work on the AOA carbon fixation cycle has been carried out with the first isolated AOA, *Nitrosopumilus maritimus* (15–20), which was isolated from a marine aquarium and represents the family *Nitrosopumilaceae* (21) (GTDB taxonomy [22], used throughout). Since then, genome comparisons have found that this cycle is also present in the first soil isolate, *Nitrososphaera viennensis* (23), and is conserved in all AOA (11, 24).

While the carbon fixation pathway within AOA has been identified, the regulation of this pathway is not well characterized. The tricarboxylic acid (TCA) cycle, gluconeogenesis, and non-oxidative pentose phosphate pathway have all been identified as conserved central carbon pathways (11, 24), but the response and control of these pathways remain unclear. Considering that terrestrial organisms like *N. viennensis* experience strong fluctuations in growth substrates and variations in environmental parameters, investigations of limiting conditions by functional genomics techniques can give insights into their adaptive capacity. To this end, inorganic carbon was chosen as a limiting substrate in this study. Unlike nitrogen and oxygen, inorganic carbon is not directly tied to the amount of energy available to AOA. It also provides a way to investigate what pathways and proteins respond within the central carbon metabolism to maintain a flow of carbon, shedding light on starvation responses, resource allocation choices, and interconnections with other environmental response pathways that may highlight other physiological points of interest within the cell.

Knowing the primary energetic reactions that drive AOA metabolism (ammonia oxidation and oxygen reduction), an energetic model was calculated in this study based on the Thermodynamic Electron Equivalents Model 2 (25) method to estimate the amount of inorganic carbon the organism would need to effectively grow. Once calculated, *N. viennensis* was grown at five different carbon concentrations with the addition of either catalase or pyruvate to test the resulting model. Proteins and metabolites were simultaneously evaluated via a combined extraction protocol and analyzed in a systems biology framework to better understand how *N. viennensis* metabolically behaves under limited carbon supply. The results show distinct shifts in the proteome and metabolome and highlight unique and distinct roles of multiple oxidative stress proteins within the cell. These results contribute to a more complete picture of redox control and the central carbon metabolism of AOA.

## RESULTS

### Growth of *N. viennensis* agrees with thermodynamic predictions

A thermodynamic model of the ammonia-oxidizing archaeon *N. viennensis* predicted a carbon consumption of 0.064 mol of inorganic carbon per 1 mol of ammonia (0.128 mol carbon/2 mol ammonia) consumed, with an overall growth equation as follows:

$$1.411O_2 + 0.064CO_2 + NH_3 \rightarrow 0.953H_2O + 0.017C_{3.85}H_{6.69}O_{1.78}N + 0.983NO_2^- + 0.983H^+ .$$

The model takes into account physiological traits of ammonia-oxidizing archaea, including a monooxygenase reaction, estimated biomass electron equivalents, activation of carbon to acetyl-CoA, and an assumed energy transfer efficiency (see Supplemental Methods) (25, 26). The model predicts that the vast majority of electron equivalents (97.2%) are needed for energy production rather than biomass synthesis (see Supplemental Methods). *N. viennensis* was grown at varying carbon concentrations (Fig. 1A) in a closed system using either exogenous catalase or pyruvate to diminish the effect

of reactive oxygen species (ROS) (27) (Fig. 1B and C) while using nitrite production as a proxy for growth. While primarily used for ROS scavenging, the use of pyruvate additionally supplies a small amount of inorganic carbon (Fig. 1C). This trait was utilized to produce a slow release of inorganic carbon for condition E, where no inorganic carbon was added.

All cultures were inoculated with comparable amounts of cells but started with carbon concentrations ranging from 0 mM to 2 mM sodium bicarbonate and the same amount of ammonia (2 mM). Cultures incubated with inorganic carbon above the theoretical threshold of 0.128 mM grew without obvious limitations (Fig. 1D, A, C, D and G). Cultures grown below this threshold were either slightly limited (0.1 mM carbon; B and F) or extremely limited (0 mM carbon; E). In the most limited condition (E), the only available inorganic carbon came from added pyruvate that was decarboxylated in the presence of hydrogen peroxide (Fig. 1C). As ammonium was present in the medium, the limited growth was directly attributed to the lack of carbon rather than an energy limitation (Fig. S1). Decarboxylation of pyruvate was also observed to give an advantage to condition F over condition B, which only had catalase (Fig. S2; Supplemental Discussion). Carbon balances were calculated using initial and final carbon concentrations in the aqueous and gas phases through dissolved inorganic carbon measurements and gas chromatography, respectively, while also accounting for $CO_2$ released by the abiotic reaction of ROS with pyruvate in conditions G, F, and E (Fig. S3; Supplemental Discussion).

## Differential protein expression in carbon-limited conditions reveals the main anabolic routes within the cell

Proteomes of the seven different conditions were subjected to a principal component analysis (PCA; Fig. 2). From the combined data, 1,264 proteins of 3,123 predicted protein coding genes (40.47%) were identified. All conditions above the carbon threshold are grouped together on the left side of principal component one (PC1), while the carbon-limited cultures are grouped on the right side of PC1. Condition E, representing the most limited culture, was grouped the farthest from carbon-replete cultures, with conditions B and F, representing slightly limited cultures, found between these two groupings. The separation of proteomes on the PCA plot closely reflects the observed growth behavior (Fig. 1).

Although the PCA showed overall differences among the proteomes of the conditions, the most abundant proteins for each condition were largely the same. Of the most abundant proteins from each condition (126 most abundant in each condition, representing the top 10% based on total proteins identified), 100 were shared (Data set S1). The high relative abundance of these proteins across all conditions indicates that they are functionally important for the cell regardless of the limitation of carbon resources (Supplemental Discussion).

Following the principal component analysis, clustering was performed on the proteomic data to identify specific proteins associated with the various conditions, and statistical tests (either analyis of variance [ANOVA] or Kruskal-Wallis) were used to determine which proteins showed a statistical difference (adjusted $P$ value ≤ 0.05) among the tested conditions. Approximately 80.7% of the total detected proteins showed a change among conditions, with 20.3% remaining constant.

Proteins were divided into seven clusters after hierarchical clustering (Fig. 3A). The carbon-limited cluster (Fig. 3A, Cluster V) represents proteins that increased in relative abundance under extreme carbon limitation. A majority of proteins (15 out of 17) predicted to make up the 3-hydroxypropionate/4-hydroxybutyrate carbon fixation cycle were found in this cluster. Based on archaeal clusters of orthologous genes (arCOG) categories (28, 29) and a hypergeometric test of all detected proteins, the E cluster was also enriched for proteins involved in amino acid transport and metabolism (Fig. 3B). In contrast, the carbon-replete cluster (Fig. 3A, Cluster VII) representing high relative

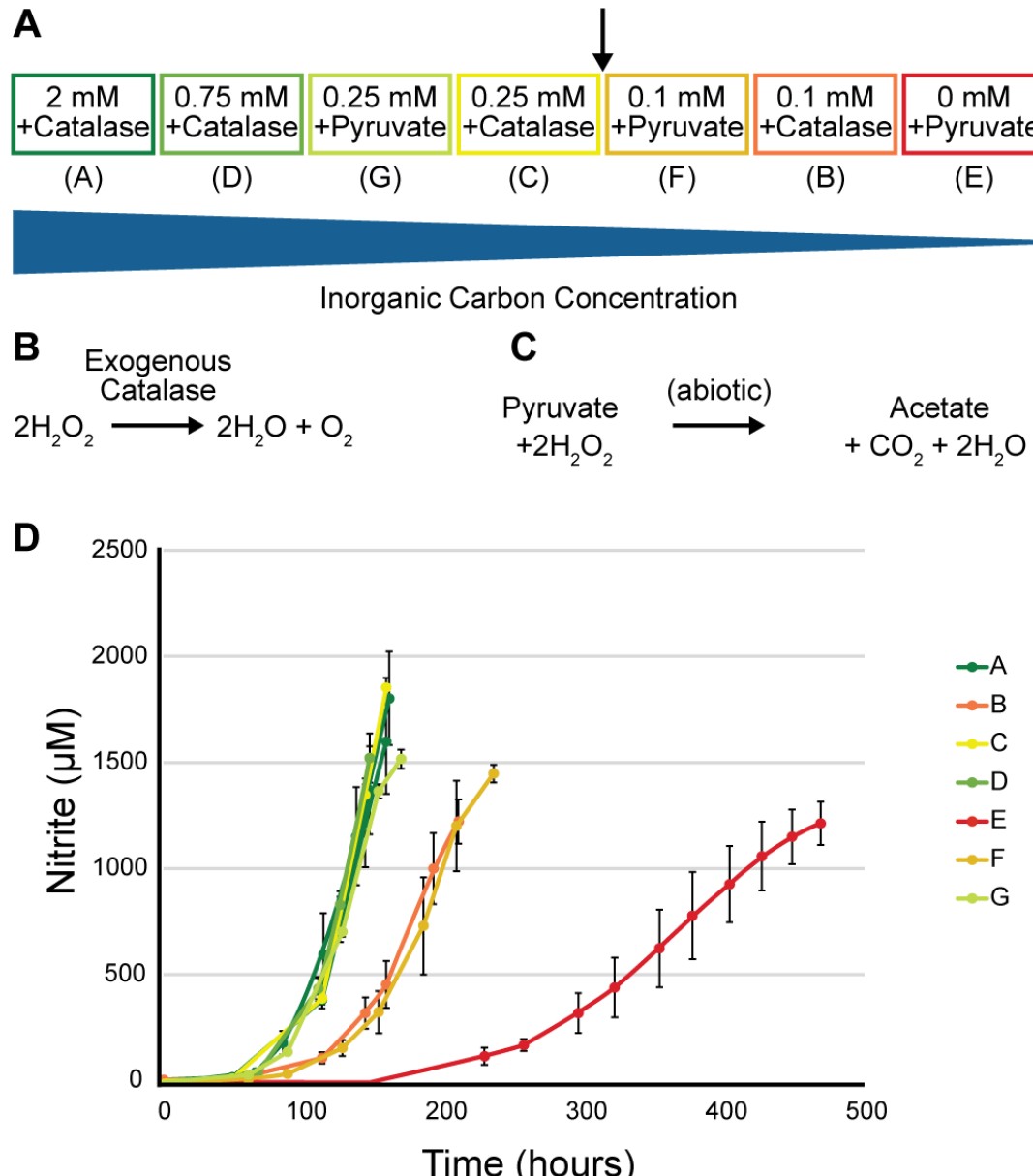

**FIG 1** Growth curves of *N. viennensis* with varying amounts of inorganic carbon. (A) Experimental design of different inorganic carbon concentrations. Black arrow indicates the calculated, theoretical carbon threshold of 0.12 mM with respect to a 2 mM ammonia concentration. (B) Reaction of catalase with hydrogen peroxide. (C) Reaction of pyruvate with hydrogen peroxide. (D) Growth curves of *N. viennensis* exhibiting responses to varying levels of inorganic carbon.

abundance in cultures that were not limited (conditions A, C, D, and G) showed enrichment for cell cycle control, transcription, and translation (Fig. 3B).

Within proteins of the 3-hydroxypropionate/4-hydroxybutyrate carbon fixation cycle, statistical tests showed that the average relative abundance (reported as label-free quantification values) for these proteins under carbon limitation (0 mM; condition E) was significantly different when compared to the standard carbon concentration (2 mM; condition A). The primary carboxylating protein, acetyl-CoA carboxylase, has been identified bioinformatically and shown to have activity in crude extracts of the marine AOA *N. maritimus* (15). A closer look revealed a strong reaction in the relative abundance of this protein across all conditions (increased abundance with less inorganic carbon), even if an effect was not observed in the growth curves (Fig. S4). With the exception of methylmalonyl-CoA mutase and methylmalonyl-CoA epimerase (MmcE;

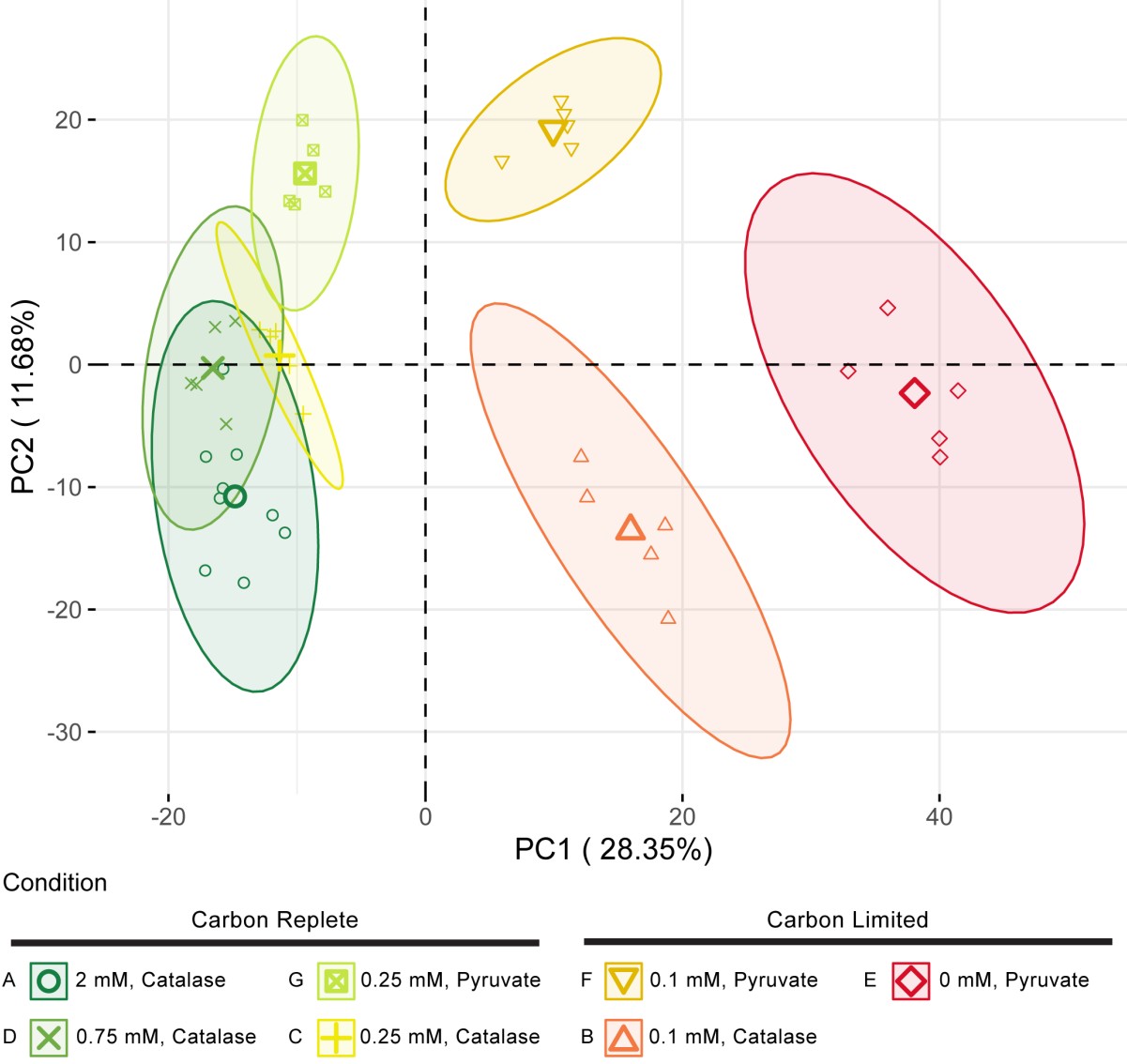

**Condition**

| Carbon Replete | | Carbon Limited | |
|---|---|---|---|
| A ⬛ 2 mM, Catalase | G ⬛ 0.25 mM, Pyruvate | F ⬛ 0.1 mM, Pyruvate | E ⬛ 0 mM, Pyruvate |
| D ⬛ 0.75 mM, Catalase | C ⬛ 0.25 mM, Catalase | B ⬛ 0.1 mM, Catalase | |

**FIG 2** PCA of *N. viennensis* proteomes. Variance within the proteomes is best described by principal component 1 (PC1) with non-limited conditions (A, D, C, and G) on the far left of PC1 and the extremely limited culture on the far right. Limited cultures B and F fall between these two groupings on PC1. Ellipses represent confidence ellipses at a level of 0.95. Centers of ellipses are marked by bold points within each ellipse, respectively.

both of which are reliably identified bioinformatically) (15), the remaining proteins can be split into those that have been biochemically characterized from *N. maritimus* (denoted with * in Fig. 4) and those that have yet to be proven but have putative candidates from previous work (denoted with ? in Fig. 4). In the case of the reduction of acryloyl-CoA and succinic semialdehyde, 10 candidate proteins had been proposed earlier based on a genomic analysis (11). The clustering analysis presented here would suggest that Adh4 (NVIE_024420) is the protein fulfilling one or both of these roles in the cycle in *N. viennensis*. The reduction of acryloyl-CoA to propionyl-CoA has been shown by the equivalent protein in *N. maritimus* (20). However, whether this same protein can also reduce succinic semialdehyde has yet to be demonstrated. If Adh4 is performing both steps, it would not be the first protein to participate in multiple steps of the cycle (i.e., Crt). In the case of GabD, Asd, and PhaAab, the proposed candidates (11) exhibit the same pattern as known proteins within the cycle. Bioinformatic predictions have identified two sets of possible genes for PhaA, NVIE_020330/ NVIE_020320 and NVIE_004940/NVIE_004950 (11, 15). Based on the proteomics results,

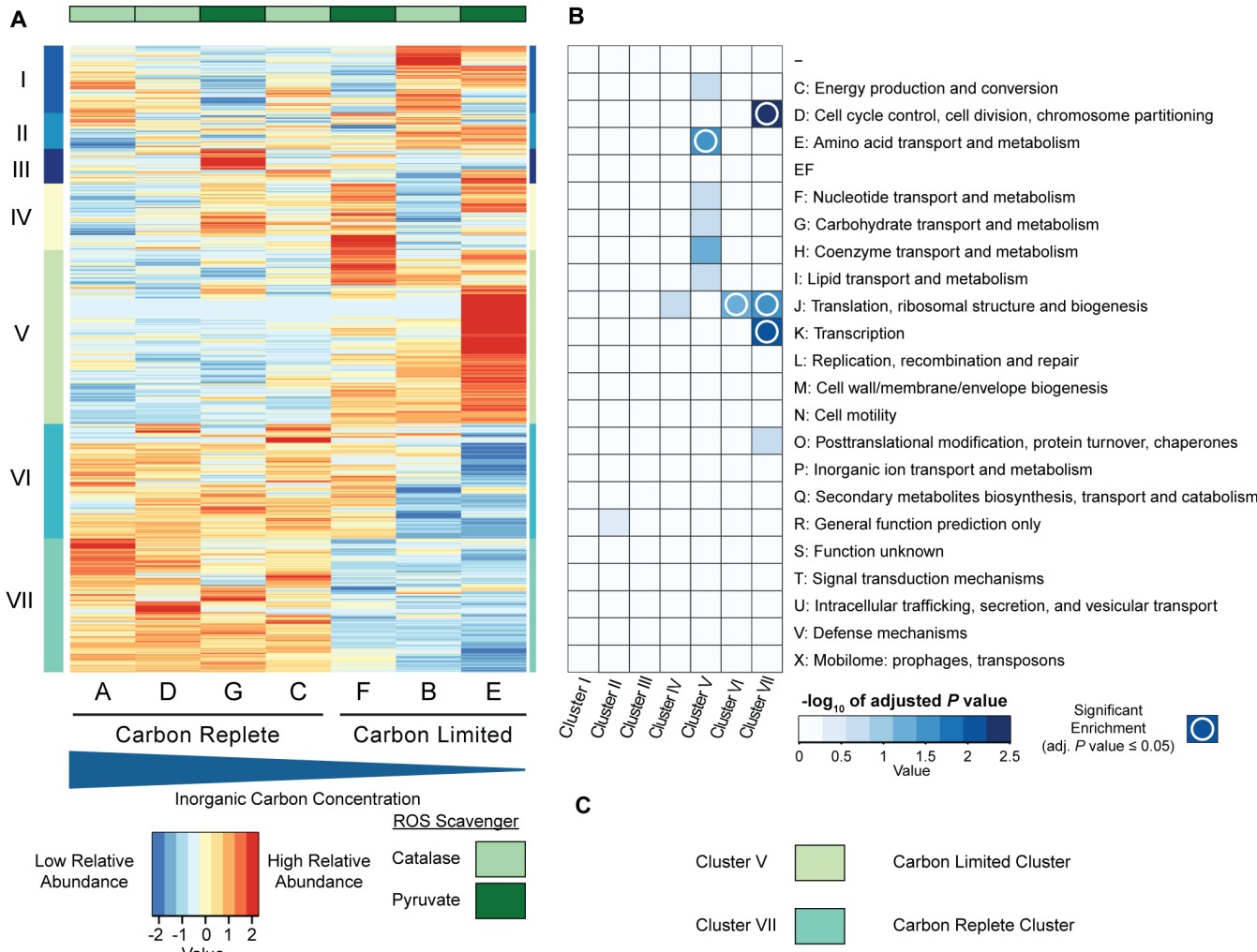

**FIG 3** Heatmap and enrichment of hierarchically clustered proteins. (A) Proteins were clustered after being centered and scaled. Clustered proteins were divided into seven clusters that are represented by the different colors and roman numerals on the left side of the heatmap. Each horizontal line represents the relative abundance of a specific protein across all conditions. Conditions are organized from highest carbon concentration to lowest carbon concentration from left to right. Color bars across the top represent conditions with either catalase (light green) or pyruvate (dark green) as an ROS scavenger. (B) Protein arCOGs enrichment analysis. Boxes with a white circle indicate arCOG categories that are enriched in their respective cluster based off of a hypergeometric test. EF represents a combined category of amino transport and metabolism (E) and nucleotide transport and metabolism (F). (C) Clusters of interest with respect to inorganic carbon abundance.

NVIE_020330/NVIE_020320 represent the thiolase participating in the cycle as they increased under carbon limitation, while NVIE_004940/NVIE_004950 were either lowly abundant or completely missing from the proteomic data (Fig. S5). As expected, proteins participating in the production (PhaE and PhaC) of the potential storage compound polyhydroxybutyrate were shown to markedly decrease in the proteome under carbon-limiting conditions (Fig. 4). The only two carbon fixation cycle proteins not found in this cluster were MmcE, which decreased under carbon limitation, and hydroxyacyl-CoA hydrogenase (PhaB), which showed some variation among conditions containing 0.1–0.25 mM of inorganic carbon rather than a strong response under limitation (Fig. S6). A summary of known and putative carbon cycle proteins can be found in Data set S1.

Other proteins within this cluster were found in the TCA cycle, gluconeogenesis, and the pentose phosphate pathway. Within the TCA cycle, proteins following the putative incorporation of succinyl-CoA were upregulated, including succinyl-CoA ligase (SucD and SucC) and subunits of succinate dehydrogenase (SdhA and SdhB). Conversely, the

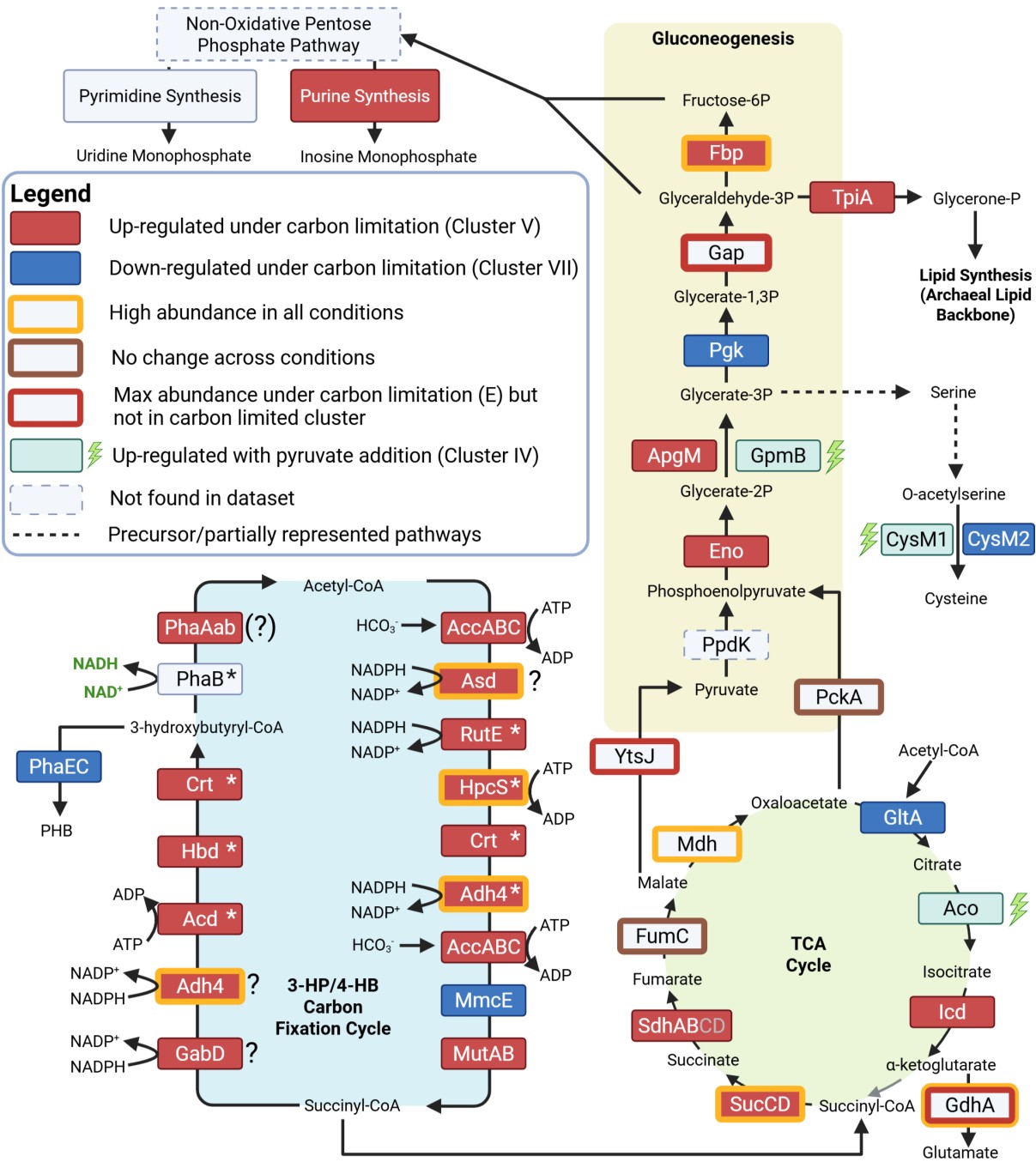

**FIG 4** *N. viennensis* core metabolism under carbon limitation. Protein abbreviations are found in rectangular boxes. Upregulated protein under carbon limitation is defined as proteins found within the carbon-limited cluster (cluster V) that show a statistical difference between condition E and condition A. Downregulated proteins under carbon limitation are defined as proteins found in the carbon-replete cluster (cluster VII) that show a statistical difference between condition E and condition A. Proteins upregulated with addition of pyruvate were found in cluster IV and showed a statistical difference between at least two conditions. A gray arrow indicates an unconfirmed step in the TCA cycle. Steps requiring NADPH, NAD+, or ATP are indicated for the carbon fixation cycle. Asterisks denote characterized proteins from AOA of the carbon fixation cycle (see Data set S1). Question marks denote steps lacking strong evidence for protein candidates (Data set S1). A question mark with brackets indicates proteins with strong bioinformatic evidence but with multiple candidates. Locus tags and accession numbers for proteins can be found in Data set S1. A carbon fixation figure including metabolites is provided in Fig. S14. Created in BioRender (L. Hodgskiss, 2025, https://BioRender.com/l00t241).

subsequent step of acetyl-CoA incorporation, citrate synthase, was downregulated under carbon limitation (Fig. 4). Fumarate hydratase exhibited no change. At the branching

point of malate, malate dehydrogenase (leading to oxaloacetate) was downregulated (cluster VI), while malic enzyme (leading to pyruvate) was upregulated (cluster III). Regardless, malate dehydrogenase maintained a higher relative abundance than malic enzyme across all conditions.

Other proteins of interest included five of six P-II regulatory proteins that were also detected with three (CnrA, NVIE_003920; CnrD, NVIE_014570; and CnrC, NVIE_014550) showing an increase under carbon limitation (cluster V) and two (GlnB NVIE_007790, cluster VI; and CnrB, NVIE_013340, cluster VII) showing a decrease under carbon limitation (Fig. S7). Additionally, in a supervised partial least squares discriminant analysis (PLS-DA), protein NVIE_010650 was found to be highly responsive to carbon limitation (Fig. S8). While no functional BLAST hits were obtained, a structural search revealed that the protein contains a putative photosynthetic reaction center domain related to potential ribosome maturation proteins (Supplemental Discussion).

## Metabolomic analysis reveals an accumulation of amino acids under carbon limitation

Selected metabolites were identified and quantified based on standard curves produced from known stock concentrations. A few of the organic acids in samples could not be accurately determined due to background contamination (citric acid, lactic acid, and oxaloacetate) and were excluded from the analysis. Pyruvate was also excluded due to its addition to conditions E, F, and G as a ROS scavenger. Metabolites that could be accurately quantified were normalized to the total amount of carbon consumed for each culture and were clustered to identify metabolites associated with specific conditions. The normalization therefore represents metabolites in terms of carbon moles of metabolite produced per mole of carbon consumed (see Supplemental Methods). A metabolite PCA plot showed a clear separation of the most limited condition (E; Fig. S9). In a clustering analysis, almost all amino acids had a higher relative abundance in the most limiting carbon condition, condition E (Fig. 5). The sugars (glucose, maltose, and trehalose) were almost exclusively associated with conditions that used catalase as a ROS scavenger (Fig. S10). Melibiose, a disaccharide, was an exception to this trend and was found predominantly in conditions that were carbon limited (Fig. S10).

## Combined analysis of proteins and metabolites identifies key cellular processes when exposed to environmental stress

Proteomic and metabolomic data were analyzed together to further investigate the metabolism of *N. viennensis*. Metabolites identified to be relatively abundant in condition E (cluster Met-IV) were analyzed with all proteins across all conditions to identify significant correlations (i.e., concurrent pattern of proteins increasing with metabolites). This revealed several proteins that consistently correlated with multiple metabolites. Within these correlations, 33 proteins were identified that correlated with five or more metabolites (adjusted $P$ value $\leq$ 0.05, $R \geq$ 0.75). The protein that most often correlated with metabolites was methionine sulfoxide reductase A (MsrA), with 15 metabolite correlations (Data set S1).

The relationship between MsrA and metabolites was further explored by correlating MsrA with all other proteins to find similar behavioral patterns. MsrA had positive correlations (adjusted $P$ value $\leq$ 0.05, $R \geq$ 0.7) with 59 proteins (Data set S1). Several proteins that correlated with MsrA also had multiple metabolite correlations ($\geq$5). This included two translationally involved proteins: the translation initiation factor eIf-2a and methionine aminopeptidase (Map; Data set S1).

MsrB2, another methionine sulfoxide reductase found in the proteome, exhibited a very different pattern compared to MsrA. It was not increased under carbon limitation but was found to be significantly higher in all conditions that were supplemented with pyruvate as a ROS scavenger rather than catalase (Fig. S11).

A correlation analysis (adjusted $P$ value $\leq$ 0.5, $R \geq$ 0.75) was also performed on all proteins with the sugars trehalose, maltose, and glucose, which appeared only under

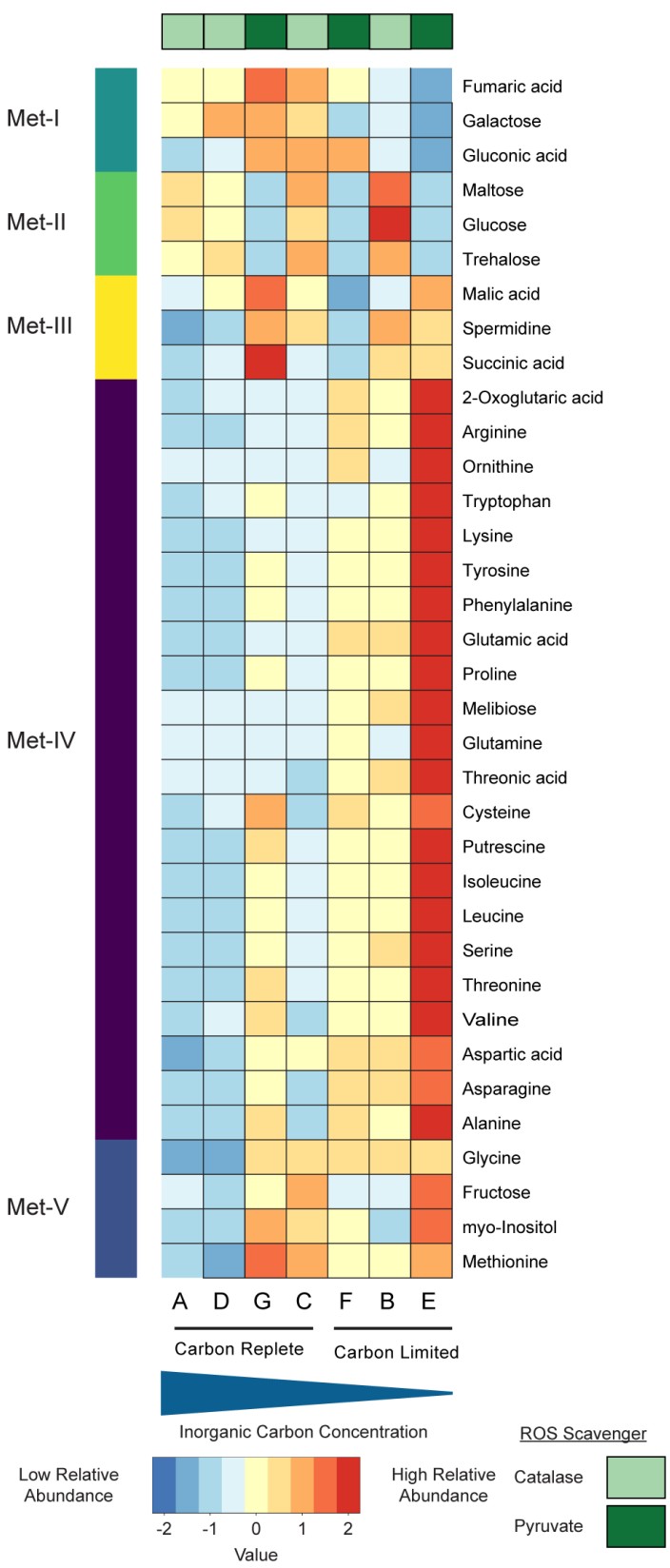

**FIG 5** Heatmap of hierarchically clustered metabolites. Metabolites were clustered after being normalized to consumed carbon and centered and scaled. Clustered metabolites were divided into five clusters that are represented by the different colors on the left side of the heatmap. Each horizontal

Fig 5 (Continued)

line represents the relative abundance of a specific metabolite across all conditions. Conditions are organized from highest carbon concentration to lowest carbon concentration from left to right. Color bars across the top represent conditions with either catalase (light green) or pyruvate (dark green) as a ROS scavenger.

conditions with catalase (Fig. S10). Five proteins were found to be correlated with all three sugars. All five had positive correlations and included ABC efflux transporter proteins (NVIE_028580 and NVIE_028560), a putative thiamine biosynthesis protein (NVIE_000430), a probable zinc uptake transporter (NVIE_021020), and a putative transcriptional regulator (NVIE_002180).

A correlation analysis (adjusted $P$ value ≤ 0.5, $R$ ≥ 0.75) with melibiose, which only appeared under carbon limitation, revealed a wide range of correlations with 72 positive and 32 negative protein correlations (Data set S1).

## Carbon metabolism and detoxification proteins responding to ROS scavenger

Due to the response of MsrB2 with respect to the choice of ROS scavenger, other known ROS detoxification proteins and core carbon metabolism proteins were manually checked for differences between the choice of ROS scavenger rather than carbon limitation. Multiple oxygen detoxification proteins and some carbon metabolism proteins were also found with MsrB2 in a protein cluster that responded specifically to the addition of pyruvate (cluster IV). In addition to MsrB2, these included an alkyl hydroperoxide reductase (Ahp2, NVIE_011770), the 2,3-bisphosphateglycerate-dependent phosphoglycerate mutase (GpmB) of the gluconeogenesis pathway, and the mitochondrial-type aconitase, Aco, albeit with less differences between conditions (Fig. S12). MsrA and TrxB were the only oxidative stress detoxification proteins found to increase under carbon limitation (cluster V). Oxidative stress detoxification proteins found to decrease under carbon limitation (cluster VII) included an alkyl hydroperoxide reductase (Ahp1, NVIE_013750), superoxide dismutase (SodA, NVIE_030260), and a putative thioredoxin (TrxX, NVIE_024000). Although decreasing under carbon limitation, SodA and Ahp1 were included with proteins that exhibited the highest relative abundance across all conditions. Two thioredoxins (TrxA1 and TrxA2, NVIE_029260 and NVIE_030000, respectively, both in cluster VI) were also found to decrease under carbon limitation (Fig. S11).

## DISCUSSION

### Thermodynamic model based on the primary metabolic reaction accurately predicts carbon consumption in ammonia oxidizers

The experimental design for carbon limitation of *N. viennensis* was guided by a thermodynamic model based on the Gibbs free energy of the electron acceptor and electron donor reactions of ammonia oxidizers calculated at standard conditions of 25°C and pH 7. Although growth conditions were slightly different from the model (42°C and pH ~7.2; see Supplemental Methods), this model was able to predict the needed amount of carbon for a respective amount of nitrogen (represented as ammonia) that is oxidized (0.06 mol C/mol N) and was used to establish starting inorganic carbon concentrations both above and below a calculated threshold. The use of pyruvate as a ROS scavenger in selected conditions (G, F, and E) allowed for an added boost of inorganic carbon as it was abiotically decarboxylated by hydrogen peroxide produced by the cells during growth.

The result of the model (0.06 mol C/mol N) falls directly between observed carbon fixation yields of marine ammonia-oxidizing archaea (0.1 mol C/mol N) and bacteria (0.045 mol C/mol N) using radioisotope measurements from Bayer et al. (30), supporting the validity of the assumptions made when creating the model (see Supplemental Methods). The differences in observed carbon fixation rates between the two nitrifying

clades can be attributed to the different carbon fixation pathways used by each clade, with AOA utilizing the more efficient 3-HP/4-HB pathway and AOB using the less efficient Calvin-Benson-Bassham cycle (15). The efficiency of these cycles is not accounted for in the TEEM model, leading to an underestimation of carbon fixation for AOA. This is also reflected in the carbon balances that show a rate of ~0.1 mol C/mol N in more balanced conditions (C and G; Fig. S3). Although the growth curves of F and B were affected by low amounts of carbon, cells were still able to fix a comparable amount of carbon per mole of ammonia oxidized. The most limited condition, E, exhibiting the most extreme limitation, is exclusively dependent on the slow release of carbon dioxide from pyruvate (Fig. 1B).

This slow release of carbon induces a shift of the proteome from informational processes toward building block synthesis (Fig. 3B) and therefore an increase in the relative abundance of proteins essential for the core carbon metabolism to make these building blocks, i.e., a shift from replication and active population growth toward what could be interpreted as a maintenance growth mode. This can also be seen by the high relative abundance of amino acids under carbon limitation (Fig. 5).

## Responses to carbon limitation indicate key enzymes and pathways in the central carbon metabolism

Previous work in AOA (11, 15), as well as isotopic studies in *Metallosphaera sedula* (31)—an archaeon utilizing the same carbon fixation pathway—has pointed toward succinyl-CoA as the connection point between the carbon fixation and the core central metabolism via the TCA cycle, with a subsequent connection to gluconeogenesis with PckA. The results shown here strongly support this flow of carbon in the cell. Most of the proteins facilitating this flow of carbon were found to increase in relative abundance under extreme limitation. Five notable exceptions occur at hydroxyacyl-CoA dehydrogenase (PhaB; carbon fixation pathway), MmcE (carbon fixation pathway), fumarate hydratase (FumC; TCA cycle), malate dehydrogenase (Mdh; TCA cycle), and phosphoenolcarboxykinase (PckA; gluconeogenesis; Fig. 4). In the case of PhaB, the protein did not exhibit a significant change in relative abundance between the two extreme conditions of this experiment (A and E, Fig. S6), indicating a tight regulation to keep it stably present within the cell. PhaB also represents a unique enzyme within the carbon fixation cycle as the only protein reducing, rather than oxidizing, an electron carrier and interacting with NAD$^+$, while most other energetic steps rely on NADPH (15, 18). In organoheterotrophs, concentrations of NAD$^+$/NADH and NADP$^+$/NADPH often regulate key catabolic and anabolic reactions, respectively, with the consumption of NADH by the ETC and production of NADPH by the oxidative pentose phosphate pathway playing key roles (32) and dictating the energetic state of the cell. As lithotrophs, AOA do not rely on the consumption of NADH by the ETC. Additionally, they lack many NADPH-producing pathways, including an oxidative pentose phosphate pathway (11). This leaves the regulation of metabolism by the consumption or production of these reducing equivalents ambiguous. Regardless, the interaction of PhaB with NAD$^+$ rather than NADPH, along with its dissimilar response to most other carbon fixation proteins, suggests that it is regulated by a stimulus other than carbon supply. Similarly, the downregulation of MmcE could be the result of a currently unknown regulatory mechanism within the cell.

Neither PckA nor FumC showed a statistically significant change in relative abundance across any carbon conditions, while Mdh, between FumC and PckA in the pathway, slightly decreased under carbon limitation (Fig. S5). Mdh converts malate to oxaloacetate, and while Mdh is seen to decrease in relative abundance under carbon limitation, it was still found to be within the 10% of proteins with the highest relative abundance in all conditions, highlighting its importance. An alternative fate for malate is the conversion to pyruvate by malic enzyme (YtsJ). From a genomic perspective, this pyruvate could enter gluconeogenesis via pyruvate phosphate dikinase (PpdK). Although PpdK has been detected in previous proteomic data sets of *N. viennensis* (11), this protein was not

detected in the data set presented here, leaving PckA as the primary connector between the TCA cycle and gluconeogenesis.

Similar to Mdh, several other proteins within the carbon metabolism were found to be some of the most highly abundant proteins regardless of carbon limitation (top 10% across all conditions, Fig. 5, outlined in gold) and are likely crucial for carbon flow in the cell. The flow of carbon within the cell is likely also controlled by the activity of PII proteins. Five of six P-II proteins were detected in the proteome with varying responses to carbon limitation (Supplemental Discussion; Fig. S7).

## Translation is a tightly controlled informational process under carbon limitation

The combined analysis of proteins and metabolites revealed additional insights into the cellular response of *N. viennensis* to carbon limitation. The protein with the highest correlations to metabolites, methionine sulfoxide reductase A (MsrA), is known to correct oxidative damage of free methionine within the cell (33). The protein with the fifth-highest correlation, a translation initiation factor (eIf-2a), is also known to interact with methionine to start translation of an mRNA sequence (34). MsrA also correlated with inosine-5'-monophosphate dehydrogenase (GuaB), the rate-limiting step of GMP synthesis (35), the precursor for GTP, and the energetic driver of the translational process (36). A protein with a putative annotation of the same step is also one of the most abundant proteins across all conditions (NVIE_023460). GMP synthesis appears to be uniquely targeted compared to other nucleotides, as most of the purine synthesis pathway—but not pyrimidine synthesis—was found within the most limited cluster (Fig. 4; Fig. S13; Data set S1), and the rate-limiting step of AMP synthesis, PurA, did not exhibit a statistically significant change between different conditions. The correlations of MsrA with eIF-2A and GuaB correspond strongly with the increase in relative abundance of amino acids of the metabolome and the enrichment of amino acid metabolism of the proteome under carbon limitation. A focus on translation is also a plausible explanation for the high responsiveness of NVIE_010650 from the PLS-DA (Supplemental Discussion). Taken together, this suggests translation as a bottleneck under carbon limitation and highlights key regulatory points fundamental in this process (Fig. 6). The cell could also be viewed as primed for translation, as almost all translation initiation factors are upregulated under carbon limitation, while translation elongation factors and ribosomal proteins tend to be downregulated. This might reflect the need of the cell to slow down its growth, cell division, and informational processes in response to its carbon-limited environment while maintaining the molecular machinery and substrates needed to engage in translation of crucial proteins. This approach aligns with a recent metatranscriptomic study in which microbial populations responded to warming temperatures by downregulating ribosomal genes while upregulating amino acid synthesis genes (37). This seemingly counterintuitive approach is speculated to be a resource allocation strategy for cells under stress.

## Melibiose as a lipid membrane stabilizer

A stark response of *N. viennensis* to carbon limitation was the presence of melibiose, a disaccharide that was not detected during growth in carbon-replete conditions. The pattern of melibiose is also distinct from other sugars (Fig. S10). Sugar metabolism within AOA is not well studied, but the investment of producing a disaccharide sugar under extreme carbon limitation suggests that it is an important molecule for the functioning of the cell under carbon limitation stress. One plausible role of melibiose could be as a polar head group to archaeal lipids. AOA are known to include a large amount (~40%) of dihexose head groups (38–41), and while the full molecular structure of these head groups is unknown, glucose and galactose, the monosaccharides that comprise melibiose, are known to be found in archaeal lipid analyses of intact polar lipids (42, 43). An increase in dihexose head groups could be beneficial for the cell by tightening the membrane (44) and preventing the escape of vital carbon metabolites. However,

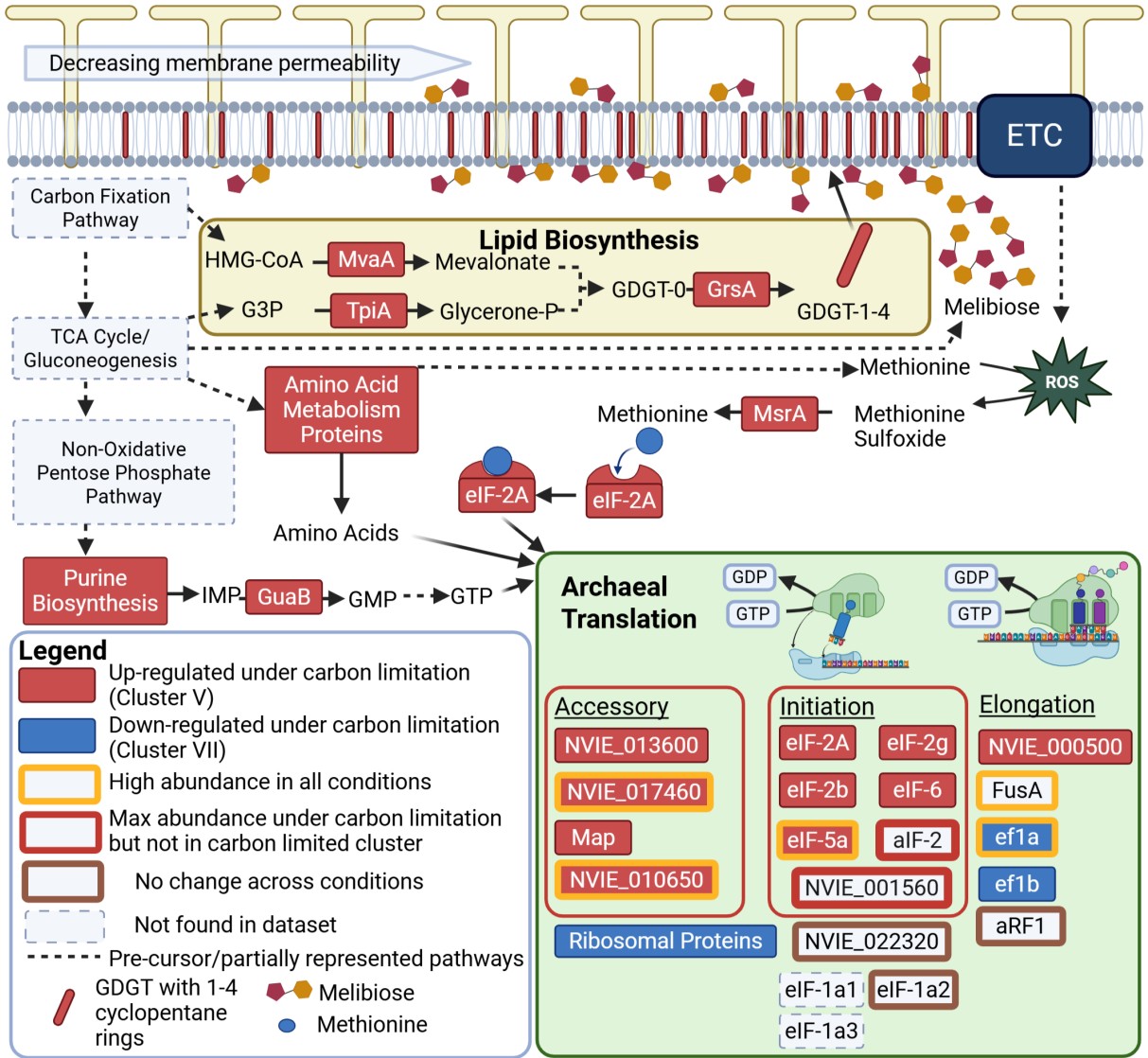

**FIG 6** *N. viennensis* targeted metabolic reactions under carbon limitation. Protein abbreviations and identifiers are found in rectangular boxes. Upregulated protein under carbon limitation is defined as proteins found within the carbon-limited cluster (cluster V) that show a statistical difference between condition E and condition A. Downregulated proteins under carbon limitation are defined as proteins found in the carbon-replete cluster (cluster VII) that show a statistical difference between condition E and condition A. Corresponding locus tags and accession numbers for proteins can be found in Data set S1. Abbreviations: ETC, electron transport chain; HMG-CoA, 3-hydroxy-3-methylglutaryl coenzyme A; G3P, glyceraldehyde-3 phosphate; glycerone-p, glycerone-phosphate; GDGT-0, glycerol dialkyl glycerol tetraether with no cyclopentane rings; GDGT-1-4, glycerol dialkyl glycerol tetraether with 1–4 cyclopentane rings, respectively; IMP, inosine monophosphate; GDP, guanosine diphosphate; GTP, guanosine triphosphate. Locus tags and accession numbers for proteins can be found in Data set S1. Created in BioRender (L. Hodgskiss, 2025, https://BioRender.com/y37o701).

currently, only the addition of monosaccharides to lipids in archaea has been documented (45, 46) rather than pre-made dihexose molecules. Alternatively, melibiose could also be interacting with the membrane via a stabilizing effect, as seen for disaccharides in other organisms (47, 48). A membrane adjustment under carbon limitation is additionally supported by an increase in the relative abundance of GrsA (NVIE_028400), one of the proteins responsible for cyclization of GDGT lipids (49) (Fig. 6). Increased cyclization of lipids in organismal membranes is known to reduce membrane fluidity (50) and has been positively correlated with reduced growth rate in the marine AOA *N. maritimus* (51, 52). AOA have been observed to release 5%–15% of their fixed carbon as dissolved organic carbon metabolites during growth (30), some of which could be escaping the

cell via passive diffusion through the membrane (53, 54). Reduced membrane fluidity would decrease permeability and likely reduce the loss of crucial carbon metabolites (particularly hydrophobic amino acids [55]) to passive diffusion of metabolites through the archaeal membrane (56).

## Choice of ROS scavenger triggers unique metabolic strategies for reactive species detoxification

Apart from carbon limitation, distinct differences in proteomes and metabolomes of *N. viennensis* were detected between growth conditions that used either pyruvate or catalase as the exogenous ROS scavenger. With pyruvate, an increase in several proteins involved in maintaining redox homeostasis was observed (Fig. 7), while with catalase, there was the exclusive presence of the sugars glucose, maltose, and trehalose (Fig. S10). While both catalase and pyruvate have been reported to detoxify ROS (27) and reactive nitrogen species (RNS) (57–59), the distinct cellular reactions can most likely be explained by the accessible sites of the ROS scavengers. Catalase constitutes a large protein that, due to its size, is not likely able to pass through the S-layer to the pseudo-periplasm where the vast majority of ROS and RNS will be produced from the electron transport chain. This likely leads to an accumulation of ROS/RNS in the pseudo-periplasm, where the critical steps of the ammonia oxidation process are likely taking place. In response to this, the production and excretion of sugars, particularly trehalose and maltose, both of which are composed of glucose, could be exported to the pseudo-periplasm to deal with this accumulation of ROS. The production of these sugars, which is only observed in the catalase conditions, would also help to relieve ROS stress within the cell (Fig. 7). This hypothesis is supported by the ROS-relieving observation of sugars in other organisms (60–62), including *N. viennensis* (albeit to a lesser extent than pyruvate; supplemental material [23, 23]), and by the correlation of all three sugars with an ABC efflux transporter protein that could transport the sugars to the pseudo-periplasm. The production of these sugars may also be connected to a putative TetR transcriptional regulator (NVIE_002180), which was also positively correlated with all three.

Conversely, most of the ROS/RNS in the pseudo-periplasm of the pyruvate-containing conditions is likely directly detoxified by pyruvate, a molecule small enough to fit through the S-layer lattice. The lack of ROS stress in the pseudo-periplasm likely negates the need for the production and excretion of sugars. While this solves the ROS problem in the pseudo-periplasm, the lack of sugar production within the cell leaves a greater amount of intracellular ROS when compared to conditions that have catalase. The increase in Ahp2 (NVIE_011770) to detoxify ROS/RNS and MsrB2 to correct damaged proteins is likely a response to this. From the data found here, it is likely that SodA and Ahp1 are the primary ROS detoxification proteins in *N. viennensis*, while Ahp2 and MsrB2 are utilized when ROS stress increases intracellularly. The strategy of keeping a primary Ahp highly expressed with additional Ahp proteins upregulated when needed is also seen in marine AOA under different ROS stresses (63). Effects within the central carbon metabolism were also found that were specific to the choice of ROS scavenger. Of particularly note is the upregulation of GpmB and CysM1 under pyruvate addition, leading to cysteine production (Fig. 4 and 7). Cysteine has been implicated in relieving intracellular ROS (64), which would align with the model presented here. While pyruvate appears to stimulate Aco in the TCA cycle, this is attributed as a response to oxidative stress and not a result of carbon assimilation via mixotrophy (Supplemental Discussion).

## An archaeal chemolithoautotrophic response to carbon limitation

As a chemolithoautotroph, the response of *N. viennensis* to carbon limitation offers a unique look at this nutrient stress without directly impacting the energy source of the cell. This is not possible in well-studied heterotrophic organisms where the carbon and energy source are directly intertwined. Even in the more similar energetic system of

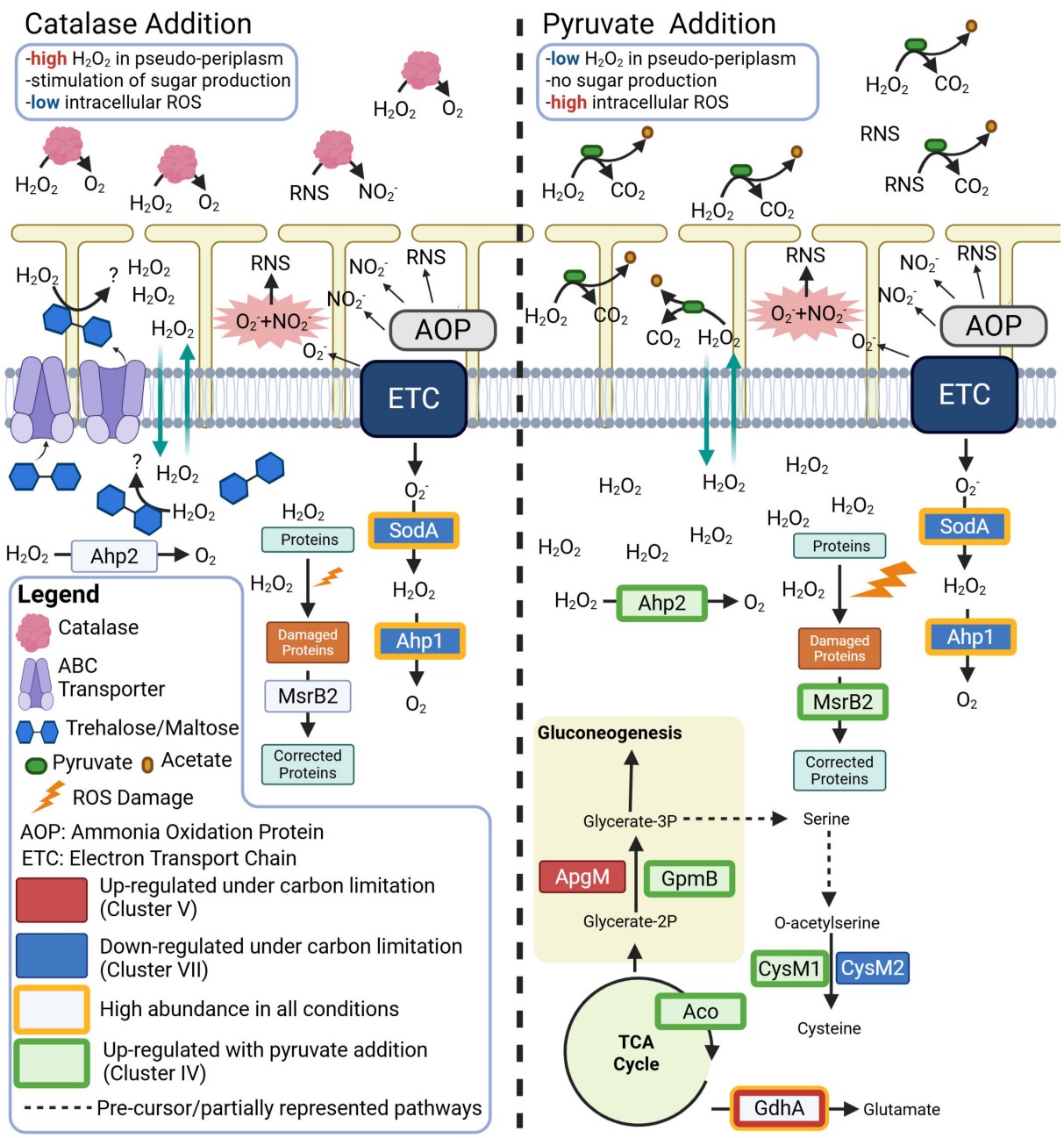

**FIG 7** *N. viennensis* model of ROS coping strategies. Protein abbreviations and identifiers are found in rectangular boxes. Upregulated protein under carbon limitation is defined as proteins found within the carbon-limited cluster (cluster V) that show a statistical difference between condition E and condition A. Downregulated proteins under carbon limitation are defined as proteins found in the carbon-replete cluster (cluster VII) that show a statistical difference between condition E and condition A. Proteins upregulated with addition of pyruvate were found in cluster (IV) and showed a statistical difference between at least two conditions. Corresponding locus tags can be found in Data set S1. Abbreviations: AOP, unknown ammonia oxidation protein(s); ETC, electron transport chain; glycerate-2P, glycerate-2 phosphate; glycerate-3P, glycerate-3 phosphate; $H_2O_2$, hydrogen peroxide; $O_2$, oxygen; $NO_2^-$, nitrite; $O_2^-$, superoxide. Locus tags and accession numbers for proteins can be found in Data set S1. Created in BioRender (L. Hodgskiss, 2025, https://BioRender.com/x36a147).

photolithotrophs, carbon limitation is usually achieved when light, and therefore energy, is not available, again directly tying carbon to the energy source of the cell. As lithoautotrophs, a decoupling of energy and carbon, and therefore the associated cellular response, is achievable in AOA. Although growth was slowed under carbon limitation, the significant reduction of nitrification observed in the cell is a direct response to carbon limitation rather than energy, as ammonia and oxygen were readily available. In

*N. viennensis*, the functional response to carbon limitation observed with this systems approach emphasized its strictly autotrophic lifestyle, as it consisted of upregulating and maintaining its standard core metabolism. This upregulation was seen even with enzymes that are already at a constitutively high level.

In addition to an observed response in core metabolism, factors involved in cellular translation, and in particular translation initiation, were associated with carbon limitation in *N. viennensis*. A regulation of translation and amino acid synthesis under carbon limitation, or nutrient stress, has been observed in species across all three domains of life. In eukaryotes, this is achieved through the highly regulated AMPK/TOR signaling system (65–67), while bacteria rely primarily on the stringent response and the regulator ppGpp (68, 69). Archaea do not have an AMPK/TOR signaling cascade, and while *N. viennensis* contains an upregulated putative SpoT/RelA gene that could initiate ppGpp synthesis, a stringent response mediated by ppGpp has not been observed in archaea. While some archaea appear to have a stringent response, it occurs without the production of ppGpp and without a drop in GTP levels (70, 71), making them unique among most bacterial stringent response systems. Although translational responses to ppGpp have since been documented in the archaeon *Sulfolobus* (now *Saccharolobus*) *solfataricus* (72), the effectiveness of ppGpp seemed to be diminished and more specialized. Regardless, specific responses in prokaryotes to carbon limitation range from breaking down existing proteins for amino acids (*Sulfolobus acidocaldarius*) (73), maintaining a ribosomal pool (*Escherichia coli*) (74), and ensuring translational fidelity (*Thermococcus kodakarensis*) (75). All of these observed responses ensure the ability of translation to proceed in the cell. In this study, *N. viennensis* was found to maintain translational activity through a concerted effort of *de novo* synthesis of amino acids and GTP, simultaneously upregulating translation initiation factors, and potentially decreasing membrane permeability to avoid loss of metabolites.

In summary, the response of *N. viennensis* to carbon limitation illustrates a tight control of translational processes as seen in other organisms across the tree of life. However, the exact regulatory mechanisms of this response remain elusive, as the typical nutrient-limitation response mechanisms in eukaryotes and bacteria are either not present or not strongly observed within the archaeal domain, likely reflective of a domain with metabolic pathways similar to bacteria but informational machinery more akin to eukaryotes. Alongside translational and core metabolism regulation, *N. viennensis* demonstrated the ability to respond to environmental stimuli in a precise manner by modifying its metabolome and/or proteome. The modeling approach combined with proteomics and metabolomics used here allowed for the detection of fine-tuned responses in *N. viennensis* that help explain its success in a wide range of soil habitats.

## MATERIALS AND METHODS

Detailed methods on the thermodynamic model development, experimental conditions, culturing of *N. viennensis*, chemical analysis (nitrite/ammonium/inorganic carbon measurements), extraction protocols (combined protein/metabolite extraction with methanol/chloroform/water), and data analysis can be found in Supplemental Methods.

## ACKNOWLEDGMENTS

We thank Lena Fragner for technical guidance during extraction of metabolites and proteins, Sonja Tischler for mass spectrometry technical support, and Hubert Kraill for assistance with running the elemental analyzer. We also thank Lisa Fürtauer, Maria Pacheco, Jakob Weiszmann, and Martin Brenner for very valuable and helpful discussions during the data analysis. We are also appreciative of Dr. Thomas Rattei and the team of the Division of Computational Systems Biology (CUBE) for providing maintenance and access to the Life Science Computer Cluster (LiSC) at the University of Vienna. This work (supplemental material) used the Water-Organic-Rock-Microbe (WORM) Portal, which is supported by National Science Foundation grants EAR-1949030 and EAR-2149016.

This project was supported by Doktoratskolleg (DK) plus Microbial nitrogen cycling—from single cells to ecosystems (Austrian Science Fund [FWF] [10.55776/W1257]), ERC Advanced Grant TACKLE (no. 695192), and the European Union's Horizon 2021–2027 research and innovation program under grant agreement no. 101079299.

L.H.H., T.N., and C.S. conceptualized and designed the experiments. L.H.H., B.B., and A.M. performed the experiments and collected data. L.H.H., M.K., Z.-H.L., B.B., A.M., W.W., and T.N. analyzed and interpreted the data. L.H.H., T.N., and C.S. wrote the manuscript. L.H.H., M.K., Z.-H.L., B.B., A.M., W.W., T.N., and C.S. reviewed and edited the manuscript.

## AUTHOR AFFILIATIONS

[1]Department of Functional and Evolutionary Ecology, Archaea Biology and Ecogenomics Unit, University of Vienna, Vienna, Austria
[2]Department of Functional and Evolutionary Ecology, Bio-Oceanography and Marine Biology Unit, University of Vienna, Vienna, Austria
[3]Center for Microbiology and Environmental Systems Science, Division of Microbial Ecology, University of Vienna, Vienna, Austria
[4]Department of Geography and Regional Research, Working Group Geoecology, University of Vienna, Vienna, Austria
[5]Department of Functional and Evolutionary Ecology, Molecular Systems Biology Unit (MOSYS), University of Vienna, Vienna, Austria
[6]Vienna Metabolomics Center (VIME), University of Vienna, Vienna, Austria
[7]Plant Evolutionary Cell Biology, Faculty of Biology, Ludwig-Maximilians-Universität Munich, Munich, Germany

## AUTHOR ORCIDs

Logan H. Hodgskiss  http://orcid.org/0000-0002-8796-3840
Melina Kerou  http://orcid.org/0000-0003-1657-3041
Zhen-Hao Luo  http://orcid.org/0000-0003-2028-8788
Barbara Bayer  http://orcid.org/0000-0003-3968-5804
Wolfram Weckwerth  http://orcid.org/0000-0002-9719-6358
Thomas Nägele  http://orcid.org/0000-0002-5896-238X
Christa Schleper  http://orcid.org/0000-0002-1918-2735

## FUNDING

| Funder | Grant(s) | Author(s) |
| --- | --- | --- |
| Austrian Science Fund | W1257 | Logan H. Hodgskiss |
| | | Melina Kerou |
| | | Zhen-Hao Luo |
| | | Christa Schleper |
| European Research Council | No. 695192 | Logan H. Hodgskiss |
| | | Melina Kerou |
| | | Zhen-Hao Luo |
| | | Christa Schleper |
| PRIORITY 'Excellent science' | No 101079299 | Logan H. Hodgskiss |
| | | Melina Kerou |
| | | Zhen-Hao Luo |
| | | Christa Schleper |

## AUTHOR CONTRIBUTIONS

Logan H. Hodgskiss, Conceptualization, Data curation, Formal analysis, Investigation, Methodology, Project administration, Visualization, Writing – original draft, Writing – review and editing | Melina Kerou, Methodology, Writing – review and editing | Zhen-Hao Luo, Formal analysis, Investigation, Writing – review and editing | Barbara Bayer, Investigation, Writing – review and editing | Andreas Maier, Investigation, Writing – review and editing | Wolfram Weckwerth, Formal analysis, Supervision, Writing – review and editing | Thomas Nägele, Conceptualization, Formal analysis, Methodology, Supervision, Writing – review and editing | Christa Schleper, Conceptualization, Funding acquisition, Methodology, Project administration, Supervision, Writing – review and editing

## DATA AVAILABILITY

The mass spectrometry proteomics data have been deposited to the ProteomeXchange Consortium via the PRIDE (76) partner repository with the data set identifier PXD060602 and 10.6019/PXD060602. Metabolomic data have been deposited to the MetaboLights database (77) under accession number MTBLS11689. Relevant scripts for data analysis can be found at the Github repository https://github.com/hodgskiss/Carbon_Limitation_Nviennensis.

## ADDITIONAL FILES

The following material is available online.

### Supplemental Material

**Data set S1 (mSystems00732-25-s0001.xlsx).** Tables S1 to S19.
**Supplemental Material (mSystems00732-25-s0002.pdf).** Supplemental text and figures.

### Open Peer Review

**PEER REVIEW HISTORY (review-history.pdf).** An accounting of the reviewer comments and feedback.

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
