## [Reviewer comments · mSystems]

Metabolic Response of a Chemolithoautotrophic Archaeon to Carbon Limitation

Logan Hodgskiss, Melina Kerou, Zhen-Hao Luo, Barbara Bayer, Andres Maier, Wolfram Weckwerth, Thomas Naegele, and Christa Schleper

Corresponding Author(s): Logan Hodgskiss, Universitat Wien

Review Timeline:

Submission Date:	May 21, 2025
Editorial Decision:	June 23, 2025
Revision Received:	August 22, 2025
Accepted:	September 8, 2025

Editor: Roland Wilhelm

Reviewer(s): The reviewers have opted to remain anonymous.

Transaction Report:

DOI: <https://doi.org/10.1128/msystems.00732-25>

Re: mSystems00732-25 (**Metabolic Response of a Chemolithoautotrophic Archaeon to Carbon Limitation**)

Dear Dr. Logan Hodgskiss:

Along with the reviewers, I found this study a compelling exploration of carbon metabolism in the lithotrophic *N. viennensis*. By decoupling C metabolism from energy generation the study provided a unique window into how carbon limitation alone shapes physiological responses. The experimental design was well executed, and the results were clearly presented. The discussion was thorough and thoughtfully engaged with all major points of interest, generating several meaningful insights. Particularly notable are the potential roles of carbon-unresponsive enzymes in the carbon cycle (e.g., PhaB), regulatory mechanisms involving P-II proteins, and stress response systems such as methionine sulfoxide reductases, all of which open new avenues for future investigation.

Thank you for submitting this high-quality research to mSystems. We welcome the publication of your manuscript following the careful consideration and response to all revisions suggested by the reviewers as well as my own comments here:

Line comments

L213: typo -> "P value < 0.5" should be "P value < 0.05"

L378: introduce all acronyms, i.e., RNS.

L439: '...decreasing membrane permeability...' <- The authors should not conclude this as a fact. This is a reasonable inference, but requires additional evidence as membrane permeability was not measured the present study. Please temper this conclusion.

Figure 1A. Clarify that the concentrations correspond to sodium bicarbonate. That it serves as the source of inorganic C is evident in the text, but is missing from the figure and the legend. Consider adding a "+" sign before the catalase and pyruvate, since one first glance the concentrations appear to correspond to these additions, rather than sodium bicarbonate.

#####

Revision Guidelines

Sincerely,
Roland Wilhelm
Editor
mSystems

Reviewer #1 (Comments for the Author):

A thorough investigation of AoA response to DIC limitation via proteomics and metabolomics. As lithotrophs, this study isolates the effect of carbon limitation from energy limitation and provides a clear indication of carbon use strategies shifting under DIC limitation.

This paper has significant results important to the field and is well written. A few notes below.

Intro

oLine 56-58: Awkward wording. Missing a word?

oLine 78: Awkward wording. Missing a word?

Supplemental Methods

oLine 42: Brings up condition F but hasn't defined it in the supp. Discussion yet

It is interesting that PC1 aligns with DIC concentration and PC2 with catalase vs pyruvate. Was pyruvate added to diminish effects of ROS (Line104) or as a small source of DIC? Or both? Was this expected/ not expected? If pyruvate as a source of CO₂ via interaction with ROS and is not the results of mixotrophy, is this a novel finding? If so, this should be discussed in the main text.

oSimilarly, how were the concentrations of catalase and pyruvate chosen?

oLine 171: Can the statement that the nitrogen source and electron donor are the same be clarified? This seems to contradict what is discussed later in the same paragraph.

oit should be confirmed that the appropriate pH and temperature would not alter the thermodynamic calculations significantly to properly evaluate the threshold reached by experiments. For example, the gibbs for the reduction of oxygen at varying conditions can be calculated readily at <https://worm-portal.asu.edu/>

•If the cell is greatly changing its proteome and amino acid budget as indicated by the proteomics, how might that change the C:N composition used for the energetics calculation?

•Figure S3 - Could the authors comment on why there would be a peak in DNA for condition C and peak in protein for G and F? Especially the peak in DNA given that is not where the biomass peak is. Why might there be more DNA but not more cells?

Results

oPlease define threshold for proteins considered high abundance.

Figure 1A - The most obvious interpretation would be the concentrations apply to the item listed below (aka 2mM of catalase) rather than 2mM DIC + Catalase. Also change "Carbon Concentration" under the blue arrow to say Inorganic Carbon Concentration to help with clarity.

Discussion

oLine 252-253: Please include that the Gibbs was calculated at 25C and pH of 7.0 and not the actual growth conditions.

oLine 296-297: Could be rephrased for clarity.

Reviewer #2 (Comments for the Author):

The study of the regulation of carbon metabolism in strict autotrophs is challenging because there are no alternative conditions besides autotrophy to analyse. In this study, the authors examine the regulation of carbon metabolism during carbon limitation and the effects of the presence of reactive oxygen species scavengers in the medium. Using metabolomics and proteomics approaches, they identified a number of proteins potentially involved in different metabolic processes in the cells, including autotrophy. The inevitable weakness of this approach is that many of the proteins have not yet been identified experimentally

and the functions of some genes have only been hypothesized. Future studies will reveal whether these assumptions are correct, but the information obtained in this study will undoubtedly aid the identification of their function in the future. There are only a few minor points that should be addressed before publication.

Please include the conditions in the (supplementary) figure legends. Otherwise, it is difficult to analyze the data. It is especially annoying in the titles of the figures (e.g., Fig. S2)

Fig. S3A: was the carbon consumption measured? Or the carbon content of the cells?

Fig. S9: "% of Carbon Consumed": do you mean % or proportion?

L. 60: „hydroxpropionate": hydroxypropionate

L. 70: succinyl-CoA

Please clearly indicate which proteins of the HP/HB cycle were experimentally proven. Please note the Nmar_1565 was identified as acryloyl-CoA reductase (<https://pubmed.ncbi.nlm.nih.gov/31188584/>), and that the biochemical characterization of Nmar_1028 (<https://pubmed.ncbi.nlm.nih.gov/34290692/>) and Nmar_1308 (<https://pubmed.ncbi.nlm.nih.gov/33472982/>) has been published

Supplementary, I 381: „H₂ N₂CO₂ (mix ratio 7:1:1)": „H₂:N₂:CO₂ (mix ratio 7:1:1)"?

Re: mSystems00732-25R1 (**Metabolic Response of a Chemolithoautotrophic Archaeon to Carbon Limitation**)

Dear Dr. Logan Hodgskiss:

Thank you for making the necessary changes. The reviewers and I believe the work is ready for publication. We thank you for choosing mSystems and for engaging in interesting scientific work.

Your manuscript has been accepted, and I am forwarding it to the ASM production staff for publication. Your paper will first be checked to make sure all elements meet the technical requirements. ASM staff will contact you if anything needs to be revised before copyediting and production can begin. Otherwise, you will be notified when your proofs are ready to be viewed.

Sincerely,
Roland Wilhelm
Editor
mSystems

Reviewer #1 (Comments for the Author):

All comments addressed.

Reviewer #2 (Comments for the Author):

The authors have properly addressed all my comments.